# Numerical Approach to the Plasmonic Enhancement of Cs_2_AgBiBr_6_ Perovskite-Based Solar Cell by Embedding Metallic Nanosphere

**DOI:** 10.3390/nano13131918

**Published:** 2023-06-23

**Authors:** Kyeong-Ho Seo, Xue Zhang, Jaehoon Park, Jin-Hyuk Bae

**Affiliations:** 1School of Electronic and Electrical Engineering, Kyungpook National University, 80 Daehakro, Bukgu, Daegu 41566, Republic of Korea; tjrudgh0826@naver.com; 2College of Ocean Science and Engineering, Shandong University of Science and Technology, Qingdao 266590, China; zhangxue00@sdust.edu.cn; 3Department of Electronic Engineering, Hallym University, Chuncheon 24252, Republic of Korea

**Keywords:** metallic nanospheres, Cs_2_AgBiBr_6_ perovskite solar cells, finite-difference time-domain method, plasmonic performance

## Abstract

Lead-free Cs_2_AgBiBr_6_ perovskites have emerged as a promising, non-toxic, and eco-friendly photovoltaic material with high structural stability and a long lifetime of carrier recombination. However, the poor-light harvesting capability of lead-free Cs_2_AgBiBr_6_ perovskites due to the large indirect band gap is a critical factor restricting the improvement of its power conversion efficiency, and little information is available about it. Therefore, this study focused on the plasmonic approach, embedded metallic nanospheres in Cs_2_AgBiBr_6_ perovskite solar cells, and quantitatively investigated their light-harvesting capability via finite-difference time-domain method. Gold and palladium were selected as metallic nanospheres and embedded in a 600 nm thick-Cs_2_AgBiBr_6_ perovskite layer-based solar cell. Performances, including short-circuit current density, were calculated by tuning the radius of metallic nanospheres. Compared to the reference devices with a short-circuit current density of 14.23 mA/cm^2^, when a gold metallic nanosphere with a radius of 140 nm was embedded, the maximum current density was improved by about 1.6 times to 22.8 mA/cm^2^. On the other hand, when a palladium metallic nanosphere with the same radius was embedded, the maximum current density was improved by about 1.8 times to 25.8 mA/cm^2^.

## 1. Introduction

In the last few decades, metal halide perovskite solar cells (PSC) have emerged as promising photovoltaic cells due to their low cost, long electron/hole diffusion length, and excellent optoelectronic properties [1,2,3]. Metal halide perovskite materials have a stoichiometric ratio of ABX_3_, where A is a monovalent organic cation, B is a divalent cation, and X is a halide anion [4,5], and most of these metal halide perovskites in the B sites contain lead (Pb^2+^), which is toxic and can be harmful to the human body. To overcome these drawbacks, non-toxic, eco-friendly halide perovskites that can replace the Pb^2+^ atom at the B site are actively being researched. Among them, Cs_2_AgBiBr_6_ perovskites have attracted interest as they are Pb^2+^ free, have high structural stability, and have unique photoelectric properties [6,7,8,9]. Additionally, long carrier recombination, lifetime, and superior defect tolerance properties make Cs_2_AgBiBr_6_ perovskites potential candidates for next-generation perovskite photovoltaics [9,10]. However, its considerably low power conversion efficiency (PCE ≈ 3%) due to a large indirect band gap and poor light-harvesting capability [11], which results in low short-circuit current density (J_sc_), hinders their commercialization.

To improve these limitations, metallization by using metallic nanoparticles (MNSs) in the optoelectronic application has been proposed as a method of improving the plasmonic phenomena and photocatalyst effect [12,13]. Embedding MNPs in the solar cell contributes to expanding the optical path length of the absorption layer and improves the light trapping capability, thus enhancing light harvesting [13,14]. To improve the light-harvesting capability, to date, research on improving performance by incorporating this metallization has been widely performed. Zhang, W. et al. synthesized Au@SiO_2_ core-shell NPs and incorporated them into a perovskite solar cell, which revealed that Au@SiO_2_ core-shell NPs reduced the exciton binding energy of the perovskite and improved the J_sc_ of the perovskite solar cell by conducting a photoluminescence study [15]. Plasmon metallic nanostructures enhance light absorption and exciton generation through localized surface plasmon resonance (LSPR). LSPR is a collective oscillation of electrons caused by the interaction between light and electrons on a metal surface. By applying this LSPR phenomenon, research on incorporating plasmonic metallic nanostructures into perovskite solar cells has been widely performed, and results are demonstrated via experimental and simulation processes. Yao, K. et al. incorporated Ag@TiO_2_ NPs into perovskite solar cells and found that when Ag@TiO_2_ NPs were embedded in the perovskite layer, they performed a strong reaction due to the LSPR effect in the specific wavelength region [16]. In addition, Wu et al. embedded Au@SiO_2_ nanorods into the perovskite solar cell and enhanced the PCE by over 40% [17].

In the use of NPs, morphological features such as size and shape play a crucial role in the performance of solar cells, such as charge transport and efficiency. Representatively, the size-dependent optical properties of NPs have been a widely discussed issue since they have a significant influence on the plasmonic performance of nanostructures. According to Chen, H. et al., the refractive index sensitivity of Au NPs with different sizes dispersed in water glycerol mixtures can be changed by changing the size of the Au NPs, indicating that the size of the NPs is a very important factor in the optical effect [18]. Additionally, Lin, G.-J. et al. embedded polystyrene nanospheres (NSs) in solar cells via simulation, controlled their diameter, and demonstrated superior light-harvesting capabilities at 450 nm diameter [19]. Therefore, embedding MNP plays a significant role in improving the PCE of solar cells. Nevertheless, the study of Cs_2_AgBiBr_6_ perovskites has, to the best of our knowledge, rarely been carried out.

Herein, the Cs_2_AgBiBr_6_ perovskite layer (600 nm)-based solar cell with embedded metallic nanospheres (MNSs) was designed with a titanium dioxide (TiO_2_) electron transport layer (100 nm) and a spiro-OMeTAD hole transport layer (200 nm) [20,21,22], and then the light-harvesting properties were investigated through a plasmonic approach [23,24] by solving the electromagnetic equations via simulation. We used the finite-difference time-domain (FDTD) method for simulating the electromagnetic distribution and absorption properties [25,26]. Au and Pd, which were regarded as noble and transition metals, respectively, were selected as MNSs, and the radius (R) of these MNSs was scaled in the range of 20–200 nm, and then J_sc_ and absorption properties were investigated [12,13,27,28].

## 2. Mathematical Algorithm

### 2.1. Fermi Golden Rule

Prior to the simulation of an MNP-embedded solar cell, it is necessary to remember Fermi’s golden rule (FGR) [29,30]. FGR equation for ordinary photo-effect description is:(1)$$𝛿\omega_{0}=\frac{4\sqrt{2}}{3}\frac{\mu^{\frac{5}{2}}e^{2}}{m_{p}^{2}\omega \varepsilon \hbar^{3}}(\frac{4\pi \varepsilon_{0}\varepsilon E_{0}^{2}V}{8\pi \hbar \omega }){\left(\hbar \omega -E_{g}\right)}^{\frac{3}{2}}$$

This equation can be transformed into a photo-effect by MNS:(2)$$𝛿\omega =\frac{4}{3}\frac{\mu^{\frac{3}{2}}\sqrt{2}\sqrt{\hbar \omega -E_{g}}e^{2}D_{0}^{2}}{a16\pi^{2}\varepsilon_{0}^{2}\varepsilon^{2}\hbar^{4}}$$

The application of Fermi’s golden rule to the plasmonic photovoltaic effect allows the conventional photovoltaic effect to be generalized by plasmon coupling with band electrons in a semiconductor substrate on which metallic nanoparticles are deposited or embedded. In this way, it was shown that metallic nanoparticles embedded in perovskite photovoltaic cells increase the overall cell efficiency through two different channels—the optical channel, due to the enhancement of photon absorption [30], and the electrical channel, due to the reduction of exciton binding energy [15,29]. In this paper, we consider only the optical plasmonic effect by numerical solution of the classically formulated problem in terms of Maxwell’s equations for a perovskite cell with embedded metallic nano-components.

### 2.2. Finite Difference Time Domain Algorithm

The FDTD method is a technique mainly used for performing calculations and interpreting information about electromagnetic waves without any space or time restrictions by applying the finite difference method to Maxwell’s equations.
(3)$$\nabla \times \overset{\to }{E}=-\frac{\partial \overset{\to }{B}}{\partial t}$$
(4)$$\nabla \times \overset{\to }{H}=\overset{\to }{J}+\frac{\partial \overset{\to }{D}}{\partial t}$$
where $\overset{\to }{E}$, $\overset{\to }{B}$, $\overset{\to }{H}$, $\overset{\to }{J}$, and $\overset{\to }{D}$ are the electric field, magnetic induction intensity, magnetic field, electric displacement, and current density, respectively. Based on Maxwell’s equations, basic two-dimensional (2-D) FDTD was introduced in this simulation [30].

#### 2.2.1. Two Differential Finite Difference Time Domain

From Maxwell’s equations,
(5)$$\frac{\partial \tilde{D}}{\partial t}=\frac{1}{\sqrt{\varepsilon_{0}\mu_{0}}}\cdot \nabla \times H$$
(6)$$\tilde{D}\left(\omega \right)={\;\varepsilon }_{r}^{*}\left(\omega \right)\cdot \tilde{E}\left(\omega \right)$$
(7)$$\frac{\partial H}{\partial t}=-\frac{1}{\sqrt{\varepsilon_{0}\mu_{0}}}\nabla \times \tilde{E}$$

The transverse-magnetic (TM z-polarized) mode equations and transverse-electric (TE z-polarized) mode equations contain ${\tilde{E}}_{y}$ and ${\tilde{H}}_{z}$. Equations ((8)–(11)) explain the TM equations. Note that ‘~’ over the E and D fields has been dropped [31].
(8)$$\frac{\partial D_{z}}{\partial t}=\frac{1}{\sqrt{\varepsilon_{0}\mu_{0}}}\left(\frac{\partial H_{y}}{\partial x}-\frac{\partial H_{x}}{\partial y}\right)$$
(9)$$D\left(\omega \right)=\varepsilon_{r}^{*}\left(\omega \right)\cdot E\left(\omega \right)$$
(10)$$\frac{\partial H_{x}}{\partial t}=-\frac{1}{\sqrt{\varepsilon_{0}\mu_{0}}}\frac{\partial E_{z}}{\partial y}$$
(11)$$\frac{\partial H_{y}}{\partial t}=\frac{1}{\sqrt{\varepsilon_{0}\mu_{0}}}\frac{\partial E_{x}}{\partial x}\;\;$$

The finite-difference equations are as follows: $$\frac{D_{z}^{n+\frac{1}{2}}\left(i,j\right)-D_{z}^{n-\frac{1}{2}}\left(i,j\right)}{\Delta t}=\frac{1}{\sqrt{\varepsilon_{0}\mu_{0}}}\left(\frac{H_{y}^{n}\left(i+\frac{1}{2},j\right)-H_{y}^{n}\left(i-\frac{1}{2},j\right)}{\Delta x}\right)-\frac{1}{\sqrt{\varepsilon_{0}\mu_{0}}}\left(\frac{H_{x}^{n}\left(i,j+\frac{1}{2}\right)-H_{y}^{n}\left(i,j-\frac{1}{2}\right)}{\Delta x}\right)$$
$$\frac{H_{y}^{n+1}\left(i,j+\frac{1}{2}\right)-H_{z}^{n}\left(i,j+\frac{1}{2}\right)}{\Delta t}=-\frac{1}{\sqrt{\varepsilon_{0}\mu_{0}}}\left(\frac{E_{z}^{n+\frac{1}{2}}\left(i,j+1\right)-E_{z}^{n+\frac{1}{2}}\left(i,j\right)}{\Delta x}\right)$$
(12)$$\frac{H_{y}^{n+1}\left(i+\frac{1}{2},j\right)-H_{y}^{n}\left(i+\frac{1}{2},j\right)}{\Delta t}=\frac{1}{\sqrt{\varepsilon_{0}\mu_{0}}}\left(\frac{E_{z}^{n+\frac{1}{2}}\left(i+1,j\right)-E_{z}^{n+\frac{1}{2}}\left(i,j\right)}{\Delta x}\right)$$

The systematic interleaving of fields is shown in Figure 1.

#### 2.2.2. Numerical Plane Wave in Finite Difference Time

A plane wave can be derived by evoking the properties of an analytic plane wave [32]. Consider Maxwell’s equations for the $\vec{E}$ and $\vec{H}$ in Cartesian coordinates $\vec{x}=\left(x,y,z\right)$. Considering $\vec{P}\;\equiv \left(P_{x},P_{y},P_{z}\right)=\left(\sin\theta \cos\emptyset ,\;\sin\theta \sin\emptyset ,\cos\theta \right)$ as a unit vector that defines the normal direction of a planar wavefront referenced against the Cartesian $\vec{x}$ system, where θ and ∅ are the polar and azimuthal angle, respectively, the electromagnetic fields for a plane wave that propagates along the direction are of the form $\vec{E}\left(r,t\right)$ and $\vec{H}\left(r,t\right)$, where the scalar r is measured along a line parallel to $\vec{P}$. Based on the former formulas, the plane wave is expressed as:$$\vec{P}\times \partial_{r}\vec{E}\left(r,t\right)=-\mu \partial_{t}\vec{H}\left(r,t\right)+{\;\sigma }^{*}\vec{H}(r,t)$$
(13)$$\vec{P}\times \partial_{r}\vec{H}\left(r,t\right)=\epsilon \partial_{t}\vec{E}\left(r,t\right)+\;\sigma \vec{E}(r,t)$$
where ∂_i_ is a partial derivative with respect to i, and i is the subscript for r and t. It should be noted that the boundary condition and excitation sources are not included in the above equations but only induce to take a specific formation of the plane wave. The r and coordinate systems are related by the normal formation of the equation (r = $p_{x}x+{\;p}_{y}y+{\;p}_{z}z$) because the wavefront is a plane wave, where r is the shortest distance from the ${\vec{x}}_{0}=\left(0,\;0,\;0\right)$ position to the planar wavefront. Furthermore, one can convert (13) into a second-order partial differential formation, $\mu \varepsilon \partial_{r}^{2}E_{i}\left(r,t\right)+\;\sigma \partial_{r}E_{i}\left(r,t\right)=\partial_{t}^{2}E_{i}\left(r,t\right)$, where the subscript i ∈ {x,y,z} assigns the Cartesian component. These equations indicate that a uniform vector equation can always be expressed as a scalar equation.

#### 2.2.3. Periodic Boundary Condition

The interaction between the periodic structure and the field is described by Bloch’s theorem [33].
(14)$$\tilde{U}\left(x,y,z+d_{z}\right)=\tilde{U}\left(x,y,z\right)e^{-{\mathrm{jk}}_{z}d_{z}}$$

Equation (14) indicates a field $\tilde{U}$ propagating along a periodic structure in the z direction with periodic d_z_ with mathematical input. Where k_zd_ describes the change in magnitude and phase between successive periods and the tilde $\tilde{U}$ represents the phasor form of fields. $\tilde{U}\left(x,y,z+d_{z}\right)=\tilde{U}\left(x,y,z\right)e^{-{\mathrm{jk}}_{z}d_{z}}$ is periodic in z (with a periodic d_z_) and admits a Fourier series expansion. The interaction between the periodic structure and the field is described by Bloch’s theorem [33].
(15)$${\tilde{U}}_{p}\left(x,y,z+d_{z}\right)=\sum_{n=-\infty }^{\infty }{\tilde{A}}_{n}\left(x,y\right)e^{-2\pi \mathrm{jnz}/d_{z}}$$

$\tilde{U}$(x,y,z) can be represented as the sum of traveling waves, with k_zn_ = k_z_ + 2πn/dz in the periodic direction. Thus, the constant k_zn_ corresponds to the z of the wavenumber of the plane wave. Likewise, Bloch’s theorem describes a field $\tilde{U}$ propagating in a periodic structure. Applied to the 3D periodic structure, it can be defined as $n_{x}d_{x}\hat{x}+{\;n}_{y}d_{y}\hat{y}+{\;n}_{z}d_{z}\hat{z}$, where $d_{x}$, $d_{y}$, $d_{z}$ are the periods in the x, y, and z directions, respectively, and n_x_, n_y_, and n_z_ are integers.
(16)$$\tilde{U}\left(r+d\right)=\tilde{U}\left(r\right)e^{-\mathrm{jk}\cdot d}$$
where k is the reciprocal (Bloch) wave vector $k=k_{x}\hat{x}+{\;k}_{y}\hat{y}+{\;k}_{z}\hat{z}$.
(17)$$\tilde{U}\left(r\right)=\sum_{l,m,n}{\tilde{A}}_{l,m,n}\exp\left(-{\mathrm{jk}}_{l,m,n}\cdot r\right)$$

Equation (17) is the explanation of the $\tilde{U}\left(r\right)$ field, where k_l,m,n_ = k_xl_ + k_ym_ + k_zn_ is the reciprocal wave vector.

### 2.3. Device Simulation Algorithm

Figure 2 shows the schematic diagram of a designed solar cell. Based on the device structure designed as in Figure 2a, the simulation conditions were set as shown in Figure 2b using the Lumerical FDTD solution. Both Au and Pd were embedded in the Cs_2_AgBiBr_6_ perovskite layer, and a plane-wave source illumination with a wavelength range of 400–900 nm was set on the FTO electrode with the y-axis backward direction. The complex refractive index of Au and Pd was taken from Johnson and Christy [34,35]. To simulate the MNS, two perfectly matched layers were set, with one layer formed below the bottom electrode and the other above. Along the x-axis, the periodic boundary conditions were set according to the direction of the light. All layers were fixed, but the R of the MNSs in the Cs_2_AgBiBr_6_ perovskite layer was scaled from 20 to 200 nm in 20 nm increments. A solar generation analysis system was applied within the FDTD simulation to calculate the electrical and optical properties, including J_sc_ and absorptance. Additionally, the reflectance and transmittance were measured by forming the frequency-domain field and power monitors with linear-x types, which were formed above and below the device, respectively. Furthermore, the frequency–domain field and power monitor with 2D Z-normal monitor type were set for calculating the electric field intensity (E^2^) of the designed device. The total absorption A(λ) of the designed device is given as follows:A(λ) = 1 − R(λ) − T(λ)(18)
where R(λ) is reflectance and T(λ) is transmittance. Moreover, the absorption per unit volume of the designed device can be calculated by using Equation (19).
P_abs_ = −0.5 ω|E|^2^ imag(ε)(19)
where ω is the angular frequency, |E|^2^ is the electric field intensity, and imag(ε) is the imaginary part of the permittivity, respectively. To calculate the J_sc_ of the designed device, we used Equation (20).
(20)$$J_{\mathrm{sc}}=\frac{e}{\mathrm{hc}}\int_{400\;\mathrm{nm}}^{900\;\mathrm{nm}}\mathrm{QE}\left(\lambda \right)I_{\mathrm{AM}1.5\;G}\left(\lambda \right)d\lambda$$
where e is the charge electron, QE(λ) is the quantum efficiency, and $I_{\mathrm{AM}1.5\;G}\left(\lambda \right)$ is the AM 1.5 G solar spectrum irradiance. The J_sc_ measured here represents the maximum J_sc_ value when the number of absorbed photons is assumed to be 100%.

## 3. Result and Discussion

### 3.1. Electrical Characteristics Analysis of MNSs Embedded Device

MNS optimization was performed by scaling R from 20 to 200 nm in 20 nm increments and fixing the Cs_2_AgBiBr_6_ perovskite layer at 600 nm. The 600 nm thickness perovskite layer provides sufficient space to scale the R of the MNS up to 200 nm. According to Vincent, P. et al., the J_sc_ value at this time becomes the maximum value assuming the number of absorbed photons is 100%; thus, we depicted the J_sc_ value as J_sc,max_ [36].

As shown in Figure 3, J_sc,max_ was 14.23 mA/cm^2^ when there were no MNSs. By embedding MNSs and scaling R, J_sc,max_ showed a tendency to gradually increase and become saturated. When Au MNS was embedded and R was scaled, J_sc,max_ continued to rise until R was 60 nm and then became saturated at a value of ~20 mA/cm^2^. When the R of MNS was 140 nm, J_sc,max_ rapidly rises to 22.84 mA/cm^2^ (highest J_sc,max_) and then saturates to about 20 mA/cm^2^. For the Pd-embedded solar cell when R was 20 nm, a lower J_sc,max_ than that of a Au-embedded solar cell under the same conditions was calculated. As the R of Pd increased from 40 nm, J_sc,max_ rapidly rose and gradually saturated at ~25 mA/cm^2^. When R was 140 and 160 nm, the J_sc,max_ was >25.5 mA/cm^2^, and when R was 160 nm, the J_sc,max_ was the highest (25.9 mA/cm^2^). The reason for the vibration phenomenon of J_sc,max_ can be inferred from the scattering within the MNS embedded layer [37]. We have summarized the results in Table 1.

In general, in terms of plasmonic resonance, for example, Au, when R is less than ca. 100 nm, it showed a good agreement with the dipole approximation range [38]. Depending on the size of the MNPs, the radiation of plasmons can be different in terms of Lorentz friction, and the radiation of plasmons becomes extreme when R is less than 100 nm. Based on the complex refractive indices of Au and Pd, these facts emphasize that scaling the R of Au more than 100 nm is meaningful in this simulation for improving the light-harvesting capability of the device. 

### 3.2. Optical Characteristics Analysis of MNSs Embedded Device

The optical properties of the designed device were analyzed to determine the cause of the J_sc,max_ variation. 

Figure 4 shows the absorptance and depicts how the MNS radius affects the light-harvesting capability of the Cs_2_AgBiBr_6_ perovskite solar cell. The MNS with an R of 40, 120, and 140 nm, which yielded the minimum, average, and maximum values of the J_sc,max_ for both Au and Pd, were selected as analysis targets. As shown in Figure 4, the absorptance spectra of all MNS-embedded devices, including the reference device, show a fluctuating absorptance spectrum caused by light of a specific wavelength reflected from the Au electrode [39]. The Au electrode acts as a back reflector: the interference of light is induced in the active layer, then absorption peaks appear in several wavelength regions. By embedding plasmonic nanostructures and scaling the size, absorption enhancement can occur and is related to the light scattering caused by the LSPR. When Au MNS with an R of 40 nm were embedded, a modest rise in J_sc,max_ compared to the reference device was observed. When Au MNS with an R of 120 nm were embedded, the absorptance spectra increased further. Furthermore, analyzing the green spectra with an R of 120 nm MNS-embedded devices in detail, it was observed that the Pd embedment spectrum is slightly higher than that of the Au embedment spectrum. These results coincide with the result that the absorption capability, depending on the wavelength of the perovskite, is broadened when MNP is added to the actual perovskite layer [38,39]. Furthermore, in the blue spectrum indicating the absorptance of the solar cells with embedded MNS with an R of 140 nm, it was observed that a more enhanced absorption peak appeared in the 700–900 nm wavelength region. It was confirmed through the blue spectra that the Pd-embedded device shows at a higher J_sc,max_ because the region covering the 700–900 nm wavelength range is more broad compared to the Au-embedded device. Therefore, for larger radii, the absorptance spectra in the 400–600 nm region were similar for both Pd- and Au-embedded devices but showed a marked difference in the 700–900 nm wavelength region [40,41]. The enhancement in absorptance rate according to the MNS embedment is numerically summarized in Table 2.

Based on the reference without an embedded MNS, when designing a device with a different R of Au and Pd NS, there was a clear difference in the degree of enhancement in the absorption rate, accordingly. Focusing on MNS with an R of 140 nm, which calculated the maximum J_sc,max_, the Au and Pd embedment contributed about 30% and 40% enhancement, respectively, compared to the absorptance of the reference. While measuring the absorptance of the device, we measured the transmittance and reflectance of the device when embedding the MNSs in the solar cell. In the case of transmittance, the reference, Au-embedded, and Pd-embedded solar cells all showed values converging to zero. On the contrary, the reflectance showed a different tendency from the transmittance, as shown in Figure 5. 

The reflectance of reference for the Au- and Pd-embedded devices is close to 0.1, with slight fluctuations in the 400–477 nm range, indicating that all devices generally trap photons in this wavelength range. These results are related to the absorptance shown in Figure 3. A difference in reflectance begins to appear below 477 nm, and the reflectance spectra of each device have different characteristics above 477 nm. In the case of the reference device, the reflectance spectrum increased with peak fluctuation, and reflectance became dominant above 700 nm. For the Au-embedded device, a reflectance above 477 nm tends to be lower than that of the reference device, and for the case of the Pd-embedded device, the degree of fluctuation in reflectance was lower compared to the reference device, and the reflectance was relatively uniform and showed low values. Thus, MNS embedment was found to cause reflectance loss in the solar cell, and the loss was different depending on the type of metal, even at the same R [42]. We summarized the reflection loss result in Table 3. 

Based on the R of all MNS at 140 nm, the degree of reflection loss of Au and Pd was about 42% and 57%, respectively, showing a correlation with the absorptance of the designed devices.

### 3.3. Spatial Distribution Analysis

For a more precise analysis of the light-harvesting efficiency of the device, the P_abs_ and |E|^2^ of the device were analyzed according to the MNS embedment in the absorption layer. Figure 6 shows the xy cross-sectional view of the P_abs_ of the absorption layer. The xy cross-sectional view profile was analyzed focusing on three wavelengths, 714, 778, and 855 nm, which showed absorptance peaks above 700 nm. The red and blue shades indicate higher and lower intensities of the parameters, respectively [43]. 

As shown in Figure 6a,d,g, all reference devices showed a flat irradiance when illuminated with 714, 778, and 855 nm monochromatic light, and it varied with the absorptance. When Au is embedded, as shown in Figure 6b,e,h, it can be confirmed that the absorption rate is generally increased by ~100 times compared to a reference device. Au MNS contributes to increased absorption by inducing light trapping [14]. When Pd NSs with the same R were embedded instead of Au, the absorption area became more dominant and thicker with Au NSs, and an absorption rate of 855 nm monochromatic light was maintained at 714 and 778 nm monochromatic light. This can be confirmed with a relatively high extinction coefficient compared to Au, even at 700 nm, because of the absorption by Pd NSs with a radius of 140 nm [34]. For a precise analysis of the plasmonic resonance inside the device, the spatial |E|^2^ distribution in the absorption layer was calculated based on the P_abs_ data. Figure 7 displays the xy cross-sectional view of the spatial normalized |E|^2^ distribution of the designed device under 714, 778, and 855 nm monochromatic light. 

Like the P_abs_ measurement, the normalized |E|^2^ distribution was also calculated with the xy cross-section view profile. These |E|^2^ distribution results revealed the electric force in the absorption layer of the designed device and confirmed the change in the electric field distribution when embedded with the MNS. Compared to the reference absorption layer, the electric field distribution is more concentrated at the center around the MNS in the Au MNS-embedded absorption layer. Contrastingly, in the Pd MNS-embedded absorption layer, the electric field was distributed and spread out of the Pd NS. These results indicate that Pd MNS was more susceptible in terms of medium optical properties than Au MNS, resulting in greater plasmonic resonance [28].

## 4. Conclusions

In this study, we designed the Cs_2_AgBiBr_6_ perovskite solar cell with an embedded MNS, scaled the R of the MNS, and investigated the plasmonic effect accordingly. When the R of both Au and Pd MNS was 140 nm, maximum values of J_sc_ were calculated at 22.8 and 25.8 mA/cm^2^, respectively, and the study was performed based on this condition. Through optical property studies, including absorptance, transmittance, and reflectance measurements, we analyzed how the R of MNS affects the optical response of Cs_2_AgBiBr_6_ perovskite solar cells. Finally, the distribution of the simulated electromagnetic waves in the designed device was confirmed using the xy cross-sectional view of Pabs and the |E|^2^ distribution in the absorption layer for 714, 778, and 855 nm monochromatic light. We hope that this numerical approach serves as a guide for improving the efficiency and light-harvesting restrictions of Cs2AgBiBr6-based perovskite solar cells.

## Figures and Tables

**Figure 1 nanomaterials-13-01918-f001:**
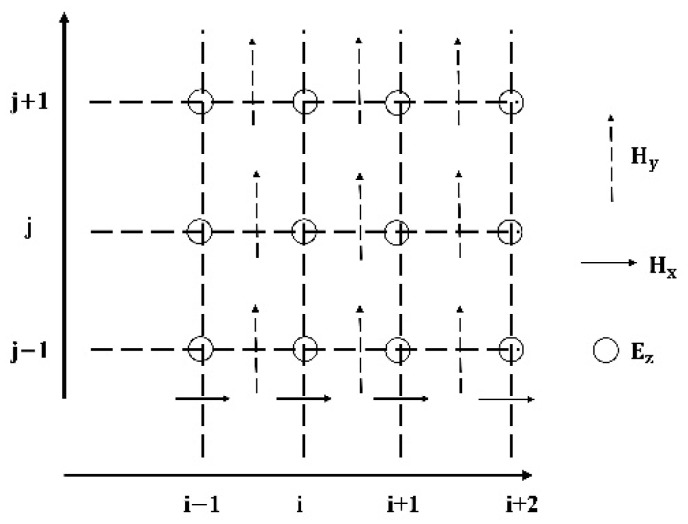
Systematic interleaving of the E and H fields for the 2D TM formulation.

**Figure 2 nanomaterials-13-01918-f002:**
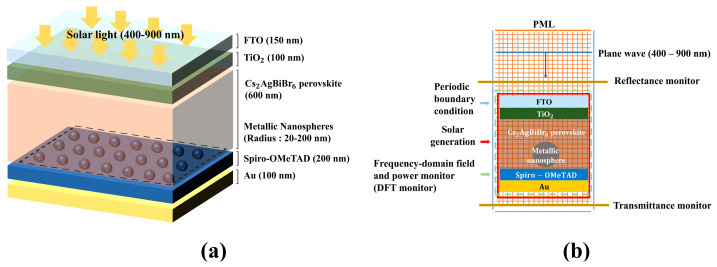
Schematic diagram of designed solar cell. (**a**) device structure. (**b**) FDTD simulation model.

**Figure 3 nanomaterials-13-01918-f003:**
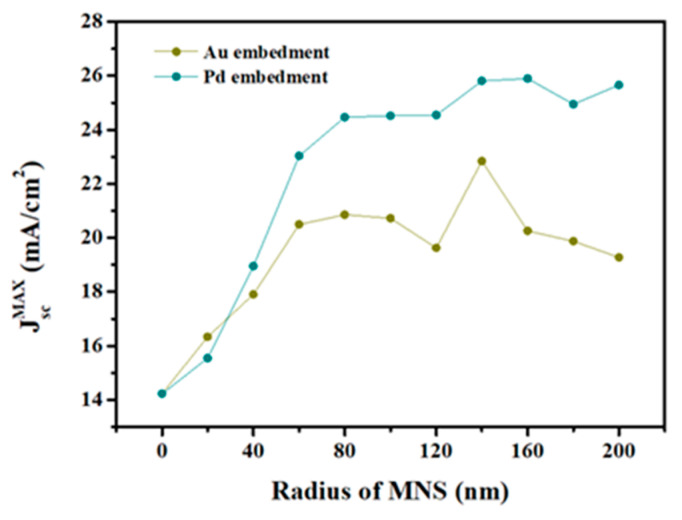
J_sc,max_ of the MNS-embedded Cs_2_AgBiBr_6_ perovskite solar cell operating under the illumination of the solar light.

**Figure 4 nanomaterials-13-01918-f004:**
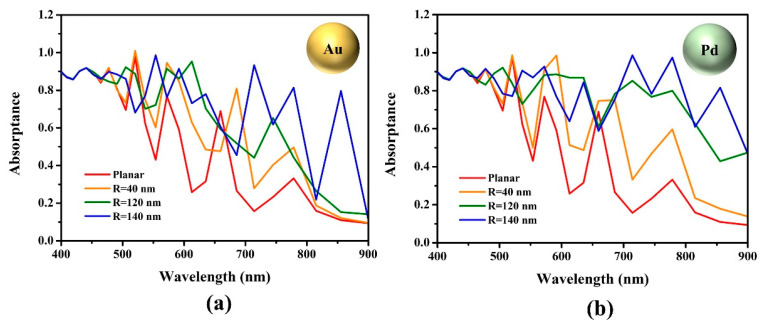
Absorptance spectra of the MNS-embedded Cs_2_AgBiBr_6_ perovskite solar cell (**a**) Au-embedded device. (**b**) Pd-embedded device.

**Figure 5 nanomaterials-13-01918-f005:**
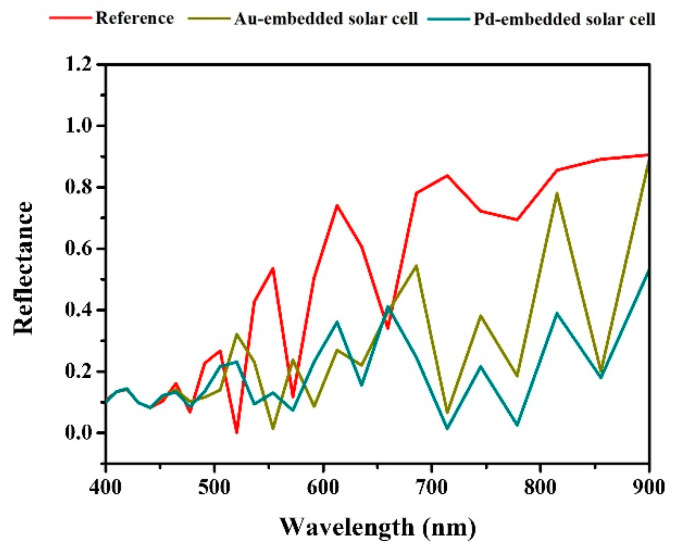
Reflectance spectra of the MNS-embedded Cs_2_AgBiBr_6_ perovskite solar cell.

**Figure 6 nanomaterials-13-01918-f006:**
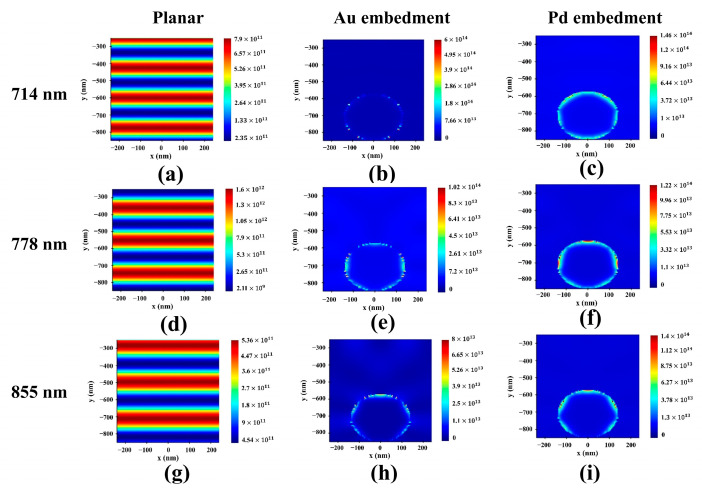
XY cross-sectional view of the P_abs_ of the absorption layer. (**a**,**d**,**g**) reference. (**b**,**e**,**h**) Au embedment. (**c**,**f**,**i**) Pd embedment.

**Figure 7 nanomaterials-13-01918-f007:**
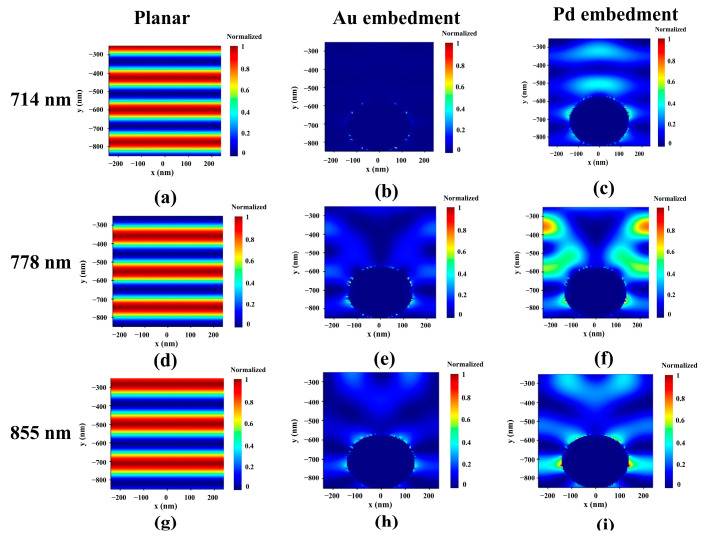
XY cross-sectional view of the normalized |E|^2^ of the absorption layer. (**a**,**d**,**g**) reference. (**b**,**e**,**h**) Au embedment. (**c**,**f**,**i**) Pd embedment.

**Table 1 nanomaterials-13-01918-t001:** J_sc,max_ for MNSs (Au, Pd)-embedded Cs_2_AgBiBr_6_ perovskite solar cell with different radii.

Radius (nm)	J_sc,max_ (mA/cm^2^)	
	Au	Pd
0	14.23	14.23
20	16.33	15.55
40	17.9	18.95
60	20.5	23.03
80	20.85	24.47
100	20.73	24.52
120	19.63	24.54
140	22.84	25.81
160	20.26	25.9
180	19.87	24.94
200	19.27	25.66

**Table 2 nanomaterials-13-01918-t002:** Absorptance performance of Cs_2_AgBiBr_6_ perovskite solar cell according to MNS embedment.

MNS Embedment	Absorptance Enhancement (%)
Without MNS (ref)	-
Au MNS	14
Au 120 nm	22
Au 140 nm	30
Pd 40 nm	17
Pd 120 nm	37
Pd 140 nm	40

**Table 3 nanomaterials-13-01918-t003:** Reflectance performance of Cs_2_AgBiBr_6_ perovskite solar cell according to MNS embedment.

MNS Embedment	Reflectance Loss (%)
Without MNS (ref)	-
Au MNS	42
Pd MNS	57

## Data Availability

Not applicable.

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
