# Peer review of "Numerical Approach to the Plasmonic Enhancement of Cs2AgBiBr6 Perovskite-Based Solar Cell by Embedding Metallic Nanosphere"

_nanomaterials, 2023, doi:10.3390/nano13131918_

Round 1
Reviewer 1 Report
The author presents The Cs2AgBiBr6 perovskite-based solar cell is embedded in a plasmonically enhanced rigorous optical model of metal nanospheres, and various plasma performance of different metal nanospheres is calculated by scaling their MNSs radii. We highly approve of the reproducibility of the experimental method designed by the authors. However, the manuscript still suffers from several problems, and we hope to receive a satisfactory answer from the authors.
1.Figure 2 is too small, Figure B text description is too much, it is recommended to be simplified, and the right 1200nm description specific value is? And is the transmission monitor placed under the simulation model Au, or is it inside? Please think more about the author.
2.There is a sentence in the text that may not be described logically clearly, please check and modify it carefully, for example, in line 56:“Among them, the size of the NPs affects the scattering of light according to Rayleigh or Mie scattering effect and affects the electric field upon light absorption. According to H. Chen, et al.”
3. In line 214 "As the perovskite layer thickness was 600 nm, it was possible to embed MNSs with a radii up to 200 nm.", what is the principle and are there any references?Why 200nm?
4. The introduction can be improved. The articles related to surface plasmon and their related applications should be added such as Micromachines 2023, 14(5), 985; Optics Express, 30(20), 35554-35566, 2022; Micromachines 2023, 14(5), 953; IEEE Photonics Technology Letters,vol.29(3), 295-298, 2017.
5. Figure 5 icon can be put into the figure and Table 1 as much as possible, it is recommended to be on one page.
Minor editing of English language required
Author Response
Thank you for your precious review.
Responses to reviews are sent as file attachments.
"Please see the attachment."
Also, we have performed the english editting project.

Reviewer 2 Report
The submission concerns an important issue of improvement of perovskite cell efficiency by metallization, especially in the case of lead free perovskite components. It is known that such a solution causes lower efficiency and its icrease via plasmonic methods is a good direction. However, the proposed approach is not fully correct -- authors named the finite element method as a ' rigorous ' -- this word should be avoided in the title. It has been verified by better insight into physics of perovskite cells that absorption increase due to plasmon effect is in perovskite cells poor and the experimentally observed increase in efficiency is caused by different quantum effect (of reducing of exciton binding energy) imposible to be accounted for by solution of macroscopic Maxwell equations, as proposed by Authors. The proposed in the paper 'optical' mathod can account only strenghtening of the local electric field close to nanoparticle curvature, which is far insufficient to describe plasmonic absorption enhacement (cf. Quantum Nano-Plasmonics, Cambridge UP 2020). In the case of perovskite cells the experimental results should be mentioned, which also clarify the situation,
Zhang et al, Nano Lett. 13 (2013) 4505.
Yao et al, ACS Nano 13 (2019) 5397.
Wu et al, J. Phys. Chem. C 120 (2016) 6996
where the insufficiency of optical approach has been experimentally pointed out and next the problem has been solved theoretically by Fermi golden rule application
Laska et al, Nano Energy 75 (2020) 104751.
This must be commented in the revised version.
Second -- Authors concluded their simulation with yhe statement that large metallic components with radius 140 nm are suitable, though their plasmon resonance study in such large metallic nanoparticles is not complete -- please refer to the literature how looks the plasmonic resonance in at least Au nanoparticles (very popular) -- at radius ca 75 nm the contribution of quadrupole starts to be important and is very large at 140 nm -- with dipole resonance red shifted -- how does it interfere with large forbidden gap in perovskite? To increase the absorption in perovskite cells a special tailoring of metallic components is needed (cf. Materials 2022, 15, 2254).
Summarizing, the paper looks rather as the study of the own approach and not of actual device. The overlap with former literature is poor. This must be improved by the discussion of metallization of perovskite cells as shown in the literature.
Moreover, the text is poorly written -- the term "method of finite elements for diffrential equation solution" should be used and not "finite difference" as used Authors, "Absorbtance" in Fig. 4 is not correct term. Additionally, the true absorption curves for Au nanoprticles are more smooth -- please refer to the Mie approach or Comsol for correspoding data. Though zig-zag curves in Fig. 4 display to some extent the tendency of true smooth contribution of quadrupole resonance and red shift od dipole one, these curves compared to known ones evidence that the presented approach is simplified even if only optical (certainly not 'rigorous').
The paper needs proof-reading (e.g., bold cannot be applied to all letters in equations).
Major revision is recommended.
medium proof-reading is recomended
Author Response

(The authors gave the same response as above.)

Round 2
Reviewer 1 Report
Accept in present form
Author Response
Thank you for your revision.
Reviewer 2 Report
Unfortunately, the authors' response is insufficient
1) The authors ignored the recommendation to add a comment about the plasmonic photovoltaic effect in perovskite cells, which is not exclusively of the absorption type (please read the previous recommendation carefully)
2) Instead the authors have added some new random off-topic references - e.g. [37] - this is clearly confusing in this case and should be avoided
3) with regard to zigzag curves in Fig. 4 - the Authors' answer was resolved by the statement that these are "fluctuations" - which fluctuations?
rather, it seems that these zigzags are the result of "fluctuations" in the method of calculation adopted
4) matters of minor importance -- Authors have retained terminology that is not commonly accepted
5) confusing bold letters are not omitted in many equations
6) the title should use "approach to" and not "on"
a more thorough revision should be repeated along all the lines suggested in the previous report, otherwise the paper cannot be accepted
moderate language verification is recommended
(and careful corrections of equations)
Author Response
We completed the answer of reviewer's comment and attached the revision letter.
"Please see the attatchment."

Round 3
Reviewer 2 Report
suggested corrections:
line 56 should be LSPR, not LSPS
in line 93 add a comment like:
“The application of Fermi golden rule to the plasmonic photovoltaic effect allows the conventional photovoltaic effect to be generalized by plasmon coupling with band electrons in a semiconductor substrate on which metallic nanoparticles are deposited or embedded in. In this way, it was shown that metallic nanoparticles embedded in perovskite photovoltaic cells increase the overall cell efficiency through two different channels -- optical one due to the enhancement of photon absorption [30] and electrical one due to the reduction of exciton binding energy [15,29]. In this paper, we consider only the optical plasmonic effect by numerical solution of the classically formulated problem in terms of Maxwell equations for a perovskite cell with embedded metallic nano-components”.
on line 194 -- "Lumerical"? is it ok?
in line 269 - in response to the allegation that the curves in Fig. 4 have a zigzag shape, Authors have added an explanation
"As shown in Figure 4, the absorption spectra of all devices with embedded MNS, including the reference device, show a variable absorption spectrum caused by light of a specific wavelength reflected from the Au electrode [39]." - this requires, however, some additional explanation (additionally MNS, or rather MNP in the above sentence)
the paper needs a thorough proofreading
English requires verification as part of a thorough proofreading of the entire text
Author Response
Thanks for detail revision.
We responsed the reviewer's comments and attached them as a word file.
"Please see the attachment."
